# A Ligand-Based Virtual Screening Method Using Direct Quantification of Generalization Ability

**DOI:** 10.3390/molecules24132414

**Published:** 2019-06-30

**Authors:** Weixing Dai, Dianjing Guo

**Affiliations:** School of Life Science and State Key Laboratory of Agrobiotechnology, G94, Science Center South Block, The Chinese University of Hong Kong, Shatin 999077, Hong Kong

**Keywords:** local beta screening, ligand-based virtual screening, machine learning, generalization ability, HIV-1 integrase

## Abstract

Machine learning plays an important role in ligand-based virtual screening. However, conventional machine learning approaches tend to be inefficient when dealing with such problems where the data are imbalanced and features describing the chemical characteristic of ligands are high-dimensional. We here describe a machine learning algorithm LBS (local beta screening) for ligand-based virtual screening. The unique characteristic of LBS is that it quantifies the generalization ability of screening directly by a refined loss function, and thus can assess the risk of over-fitting accurately and efficiently for imbalanced and high-dimensional data in ligand-based virtual screening without the help of resampling methods such as cross validation. The robustness of LBS was demonstrated by a simulation study and tests on real datasets, in which LBS outperformed conventional algorithms in terms of screening accuracy and model interpretation. LBS was then used for screening potential activators of HIV-1 integrase multimerization in an independent compound library, and the virtual screening result was experimentally validated. Of the 25 compounds tested, six were proved to be active. The most potent compound in experimental validation showed an EC_50_ value of 0.71 µM.

## 1. Introduction

Virtual screening plays an important role in drug discovery by vastly reducing the number of candidates for experimental evaluation [1,2,3]. Molecular docking is a conventional structure-based virtual screening method that optimizes the orientation of a ligand and a drug target [4,5]. However, when three-dimensional structure of the drug target is not available, ligand-based virtual screening should be the first choice for drug discovery. For ligand-based virtual screening, fingerprints with high dimension of features are firstly generated to describe the chemical characteristic of ligands, and then similarity searching or machine learning approaches are applied for screening in high-dimensional data [6,7,8]. Classification algorithms such as naive Bayes (NB) [9], K-nearest neighbors (KNN) [10], support vector machine (SVM) [11], random forest (RF) [12] and deep neural network (DNN) [13,14,15] are generally used as machine learning approaches for training a screening model, with adjustable parameters and features optimized by resampling methods such as cross-validation or bootstrapping [16,17,18,19]. However, resampling methods are usually unsatisfactory for screening because they are highly computer intensive and may lead to over-optimistic estimates due to high variance of the methods [20,21].

The objective of screening is to identify samples with high probability of positive outcome, rather than optimize a global boundary as in a classification task, and the samples to be screened usually account for only a fraction of the total samples in the positive class. Many samples in a training set may be irrelevant to the screening task and result in poor generalization ability of the model if forcibly included in resampling methods. Moreover, highly imbalanced distributions of the data (the ratio is one active compound to 1000 inactive compounds on average) [22] and high-dimensional features can aggravate the problem [23,24]. As a result, the relevant features for screening can be mocked or even undetectable in high-dimensional data if the model is trained by conventional approaches. This would lead to a poor explanation of the data and poor accuracy, which are both the concern of a screening model.

In view of such drawbacks, we here propose a new machine learning algorithm, local beta screening (LBS) to specially tackle these problems. LBS focuses locally on the samples which are important for the screening task and assesses the risk of over-fitting in an accurate and efficient way. Firstly, we established the general elements of LBS, such as loss function and the procedure for optimization. We then demonstrated the performance of LBS by a simulation study and tests on data of compound screening. Finally, we applied LBS to screen new activators of HIV-1 integrase multimerization, and the virtual screening results were experimentally validated. The open source code of LBS is available at: https://github.com/localbetascreening/lbs.

## 2. Results

### 2.1. Bias and Variance Analysis of LBS

The predictive performance of a supervised learning algorithm is not only related to the bias, but also the variance of the model. Different from conventional approaches where resampling methods are generally used for estimation of the variance, LBS includes the variance of screening in the loss function. As an example illustrated in Figure 1A,B, true positive (TP) are 2 and 6, and false positive (FP) are 0 and 1 for screening model A and B respectively. Although screening model A has less bias, the variance is more than that of screening model B. In a comprehensive view by assessing the loss function, model B performs better than A for screening.

In general, the loss function of LBS is closely related to both bias and variance of screening (Figure 1C). The lower limit of loss function is bias when variance is close to zero. When bias is more than 0.5, the value of loss function decreases with the increase of variance. However, it is totally different if bias is less than 0.5, where the value of loss function increases along with variance. This indicates that when bias is small, loss function of LBS regards variance as the dominant factor for screening automatically. Therefore, by taking both bias and variance of probability into account, loss function of LBS serves as an optimality criterion for assessing the performance of screening models.

### 2.2. A Simulation Study of LBS

To visually and intuitively demonstrate the efficiency of LBS, we tested its performance by a simulation study. A raw training set with 200 negative samples and 50 positive samples was generated from a two-dimensional uniform distribution and a normal distribution, respectively. An additional feature of random noise was then added to the raw training set, and the process was repeated 2000 times to generate multiple training sets, each of which contained one feature noise and the raw training set. Next, 2000 independent test sets were generated, each contained 800 negative and 200 positive samples from the same distribution as the training datasets for evaluating the accuracy of screening. Screening models were trained on a training set and evaluated on a test set for each time, and this was repeated to estimate the overall accuracy. During training, a model evaluated whether the additional feature was a noise or not for screening, by Equation (2) for LBS and leave-one-out cross validation for KNN and SVM. The parameters of models were optimized during training. For KNN, the number of nearest neighbors was optimized in a range of all odd numbers from 1 to 201. The polynomial kernel function was used in SVM. Degree of polynomial and the regularization parameter were optimized simultaneously by grid search, where the ranges were 1 to 5 and 500 to 1000 for degree of polynomial and the regularization parameter, respectively.

Figure 2A,B, respectively, show the value of loss function and the radius of LBS for all points in raw space without feature noise. Large value of loss function and small radius of LBS tend to share similar space, where the samples are all negative or only a few are positive. With the increase of radius, the variance of the model decreases while the bias may increase. As a result, the distribution of small loss function and large radius are irrelevant, and the optimal point (i.e., the point with minimum value of loss function) corresponds to a medial value of radius by a tradeoff between bias and variance.

The ability of a screening model to detect the noise correctly is important for both screening accuracy and model interpretation. Among the 2000 feature noise, 99.8% were successfully detected by LBS, while the ratios detected by KNN and SVM were only 92.0% and 73.2%, respectively (Figure 2C). As shown in the density profile (Figure 2D), the accuracy of screening on test sets was quite similar for KNN and SVM, while the distribution of LBS was obviously fat-tailed on the right side and its overall accuracy was significantly better than for KNN or SVM (*p* < 0.001). Therefore, LBS can assess the risk of over-fitting in a more accurate and efficient way, leading to better performance in terms of screening accuracy as well as model interpretation.

### 2.3. Application of LBS for Compound Screening in Real Datasets

In this section, we used LBS to explore real datasets and compare the performance to several classical machine learning methods for ligand-based virtual screening. The first dataset was a confirmatory biochemical assay of inhibitors of Rho kinase 2, which has previously been analyzed by several machine learning methods [25]. The second dataset was obtained from two bioassays identifying activators of HIV-1 integrase multimerization, and the performance of LBS was compared with two classical approaches for compound screening, namely NB and molecular docking. Furthermore, new compounds which might act as activators of HIV-1 integrase multimerization were screened by LBS, and the result was experimentally validated.

For the first dataset, the features were generated as previously described. Comparison of LBS to other machine learning methods described previously is illustrated in Figure 3A. Precision of LBS was 0.667 for all the first three principle components (PCs), which was higher than that of conventional approaches such as SVM, RF, J48 decision tree, and NB. Recall of LBS was 0.154 for PC1 and PC2, and it increased to 0.308 for PC3 without any loss in precision. In addition, more than 96% of the active samples were explained by nine PCs, and the number of features used in LBS was below 3% of the total features, which was significantly less than that of the other four methods (Figure 3B).

The comparison of approaches for screening of activators of HIV-1 integrase multimerization was investigated by 10-fold cross-validation, which was repeated 10 times, and the average result was used for evaluation. As for NB, different thresholds resulted in different screening accuracy. Specifically, the accuracy decreased with the increase of threshold, with a maximum accuracy of 88.9%. The threshold of LBS was optimized automatically in the training process, and the screening accuracy was 93.0% ± 2.4%, which is significantly higher than that of NB (*p* < 0.01, Figure 4A) and molecular docking (*p* < 0.01, Appendix A). Precision–recall curve (PRC) provides a global view for the results of classification (Figure 4B). As shown, the overall curves could be divided into two parts. LBS was dominant over NB for low recall, while the opposite was true for the remaining thresholds far beyond the range of LBS modeling. The area under curve (AUC) of LBS in the screened zone of PC1 (0.267 ± 0.004) was apparently larger than that of NB (0.246 ± 0.005). Surprisingly, the global AUC of LBS (0.590 ± 0.012) was even slightly larger than that of NB (0.586 ± 0.011). The balanced accuracy of LBS (56.3% ± 0.8%) was not significantly different from that of NB (56.4% ± 0.4%), and the results of Mathews correlation coefficient (MCC) were similar (0.149 ± 0.010 and 0.147 ± 0.007 for LBS and MCC, respectively). Therefore, it indicated that LBS was not only robust in the screened zone, but it also generalized well outside the screened zone.

Furthermore, permutation test was used to test the performance of LBS (Figure 4C). The values of dependent variable were randomly permutated 2000 times to calculate the loss values. The loss value of LBS obtained from the actual experiment was significantly better than from the random permutation (*p* = 1.8e^−9^).

An obvious advantage of LBS is that the result is easy for interpretation. We trained the LBS model on the whole dataset, and the optimal feature subset consisted of three features from functional-class fingerprints with a width of six bonds (FCFP_6), which are shown in Figure 5. Array of optimal identifiers are [74595001, 422052003, 1753546204], among which identifier 1753546204 was the most informative, whereas 422052003 was the second and 74595001 the least. Totally, 16 active compounds and no inactive compounds were found in the optimal screened zone by LBS (the gray-filled zone in Figure 5), and the corresponding value of loss function was 0.056. According to the permutation test conducted (Figure 4C), the screening model by LBS was significant and thus can be used as a guidance for further screening. The center was where all the three features exist, and the radius was one. It can be interpreted as a screening rule that a candidate is regarded as an active compound if its structure contains at least two of the three optimal features.

After training, we used LBS to screen potential activators of HIV-1 integrase multimerization which were not explored previously in independent compound libraries. The screening results was further validated by in vitro binding assay, and the dose-response curves of the compounds were determined. We experimentally validated the 25 compounds suggested by LBS, and the detailed dose–response data are shown in Appendix A. Among them, six were proved to be active (Table 1). The most potent compound was MCULE-2245265974 (PubChem CID 135427812), showing an EC_50_ value of 0.71 µM (Figure 6). Figure 7 illustrates the binding mode of MCULE-2245265974 to HIV-1 integrase, where we can see that MCULE-2245265974 occupies the interface between the two HIV-1 integrase monomers. The neighbors of MCULE-2245265974 in the pocket comprises residues E87, P90, Q95, E96, Y99, K103, H171, and K173. Due to the symmetry of the binding site, residues of chain A in the pocket are the same as those of chain B. This pocket at the dimer interface near K173 has been identified as a potential allosteric binding site [26,27,28]. 

## 3. Discussion

Identification of important or most informative samples in classification has been extensively studied by emphasizing samples of high variance [29] or relating the weights to loss function [30,31]. LBS provides an approach of identifying both important samples and important features for screening by defining a screened zone, whose center and radius are automatically optimized during training. During training, the screened zone is chosen so that both bias and variance of screening are minimized to ensure robustness of the model. Meanwhile, LBS focuses on the screened zone locally and neglects the samples outside, which also differs from other global approaches such as ranking methods [32,33] or one-class classification [34]. In an extreme case, LBS can be close to classification approaches when the optimized radius is large.

The principle component of LBS is similar to that of principle component analysis [35]. However, there are some differences. First, principle component of LBS is sparse, which means that each principle component of LBS is related to only a fraction of the total features. Second, as focus of screening is on the active class, we use the ratio of samples explained in active class, rather than total variance, to describe how much the data can be explained by the screening model. Priority is given to the top-ranked PCs for their high confidence level. The number of PCs can be determined by comprehensive consideration of the number of samples needed in screening and the explained ratio in the training set.

Parameter optimization for LBS is efficient because the optimality criteria is a refined form considering both bias and variance of screening, without the need of additional resampling methods such as cross validation or bootstrapping. In addition, only two parameters, center and radius of the screened zone, are required during training of LBS, and they are optimized automatically without intervention.

The feature selection in LBS is essentially a wrapper method [36], and when the number of features is huge, the optimal solution may not be ensured due to curse of dimensionality. Enlargement of the predefined parameter Ne, which is limited by the power of the computer, helps increase the probability of obtaining the optimal solution.

In summary, we propose a machine learning model for ligand-based virtual screening. The novelty of the approach is that it evaluates the generalization ability of the model in an accurate and efficient way, thus leading to better performance in terms of screening accuracy as well as model interpretation compared with that achieved by conventional algorithms for ligand-based virtual screening. Performance tests on examples and results of experimental validation indicate that our model is robust and offers a general framework for ligand-based virtual screening.

## 4. Materials and Methods

### 4.1. Local Beta Screening

The objective of LBS is to optimize screening accuracy by measuring the distance between the probability learnt from the dataset and the ideal state, and meanwhile minimize the bias and variance of screening with a refined form so that the model is robust and interpretable. The variance of screening is estimated by including the density of probability distribution in the loss function of LBS.

The ideal state of screening is that candidates are confirmed to be positive. However, the learnt probability may deviate from the ideal state. Also, variance of probability is inevitable. Suppose that a screened zone obtained is expressed as a hypersphere with center at a training sample and radius to be determined. Let *p* be the probability of samples in the screened zone belonging to the positive class. True positive, TP, and false positive, FP, are the number of positive and negative samples respectively observed in the screened zone. Then, it can be concluded that *p* conforms to beta distribution with shape parameters of TP + 1 and FP + 1 in the absence of any prior knowledge. A screening error rate can be defined as 1 – *p* to measure the closeness to the ideal state. To incorporate the influence of variance in the model, we use the expected error rate of screening as loss function of LBS:(1)$$L=\int_{0}^{1}\frac{p^{TP}{\left(1-p\right)}^{FP}}{B(TP+1,FP+1)}\cdot (1-p)dp$$
where *B* is beta function and *L* is loss function of LBS. Equation (1) can be analytically solved as:(2)$$L=\frac{FP+1}{TP+FP+2}$$

The detailed proof is shown in Appendix B. Apparently, the loss function defined above is refined and easy for optimization.

In the process of training, feature subset and screened zone are optimized by the loss function of LBS. Equation (2) is evaluated on all the samples for a given feature subset, and only the best, rather than the summation, of evaluations is used as the metric of the subset (Figure 8). The final LBS model after training consists of multiple PCs, and each PC is a sparse combination of features.

The procedure of LBS for one PC can be started by initializing a feature set with all the features in the dataset. For any given feature subset, the center and radius are optimized automatically by Equation (2). In the *i*th iteration, all the subsets containing *i* features are evaluated and then sorted by the value of loss function. If the minimum value of loss function in current iteration is not smaller than that in previous one, we update the feature set with the one in previous iteration and exit. Otherwise, we update the feature set with the first *k* unique features from the list of sorted feature subsets and go to the next iteration. The number of unique features *k* is determined as follows:(3)$$\begin{array}{l} Maximize\;\;\;\;\;k\\ subject\;to\;\;\;\;\frac{k!}{(i+1)!\left(k-i-1\right)!}\leq N_{e} \end{array}$$
where *i* is the index of iterations and Ne is the maximum number of feature subsets to be evaluated in each iteration, which is a predefined constant to control the amount of calculation. The detailed procedure of LBS for one PC is illustrated in Appendix A. After the first PC is generated, delete the samples in screening zone and repeat the procedure to generate the next PC until the maximum number of PCs or sample explained is reached.

### 4.2. Algorithm Implementation

We develop a Python implementation of LBS, which takes advantage of multiple central processing unit cores to significantly accelerate the training process. The Multinomial NB classifier [37], as implemented in the Python library “Scikit-learn”, was used for classification by NB. The package libsvm [38] was used for training of SVM, and adjustable parameters were optimized by cross validation.

### 4.3. Datasets

The dataset of biochemical assay of inhibitors of Rho kinase 2 was obtained as described in Schierz’s work [25]. The dataset for identification of activators of HIV-1 integrase multimerization was obtained from two bioassays. The inactive compounds were from a normal high-throughput count screen assay, and the active compounds were from a dose–response confirmatory assay [39,40]. Features of compounds were computed using FCFP_6 [41], as implemented within Pipeline Pilot (version 7.0.1, Accelrys, Inc., San Diego, CA, USA). The final dataset consisted of 101 active compounds and 1677 inactive compounds, described by 17764 features.

### 4.4. Molecular Docking

The sdf-formatted files containing three-dimensional coordinates of ligands were obtained from the PubChem database (National Center for Biotechnology Information, Bethesda, MD, USA) [39]. The files of structures were converted to Protein Data Bank format by Open Babel [42] (version 2.4.1, Blue Obelisk Group, San Diego, CA, USA) and then processed by MGLTools (version 1.5.6, The Scripps Research Institute, La Jolla, CA, USA). The three-dimensional coordinates of HIV-1 integrase core domain (PDB ID: 3L3V) were obtained from Protein Data Bank [43]. The ligand and water molecules in the domain were removed, and polar hydrogens were added. AutoDock Vina (version 1.1.2, The Scripps Research Institute, La Jolla, CA, USA) was used for docking [44].

### 4.5. Experimental Validation of the Screening Results by LBS

An assay was carried out as previously described to experimentally validate the compounds [45] identified by LBS as activators of HIV-1 integrase multimerization. Briefly, His-tagged and FLAG-tagged integrase (donated by professor Mamuka Kvaratskhelia at university of Colorado Denver Anschutz Medical Campus) were separately prepared with final concentration of 20 *nM*. An antibody mixture was prepared with 1.25 μg/mL of anti-6xHis-XL665 antibody (Cisbio) and 58 *ng/mL* of anti-FLAG-EuCryptate antibody (Cisbio), respectively, in Tris-HCl buffer. His-tagged and FLAG-tagged Integrase were mixed in 25 mM Tris-HCl (pH 7.4) buffer containing 150 mM NaCl, 2 mM MgCl_2_, 0.1% Nonidet P40, and 1 mg/mL BSA. 35 μL of integrase mixture was added to each well. Next, 270 nL of compounds (each in triplicate using 8-point 1:3 dilution series starting at a maximum nominal test concentration of 53.7 μM) were distributed to appropriate wells. Dimethyl sulfoxide was used as low-control, and BIB-II (donated by professor Mamuka Kvaratskhelia at university of Colorado Denver Anschutz Medical Campus) was used as high-control reference compound. After 3 h of incubation, 15 μL of antibody mixture was added to the plates. After incubation at room temperature for 2.5 h, the time-resolved fluorescence energy transfer signal was measured by the microplate reader (SpectraMax M5, Molecular Devices, LLC., San Jose, CA, USA) by exciting the plates (384 well, Corning) at 340 nm and monitoring well fluorescence at 618 nm and 671 nm with the microplate reader. All the test compounds were bought from ChemDiv (San Diego, CA, USA).

## Figures and Tables

**Figure 1 molecules-24-02414-f001:**
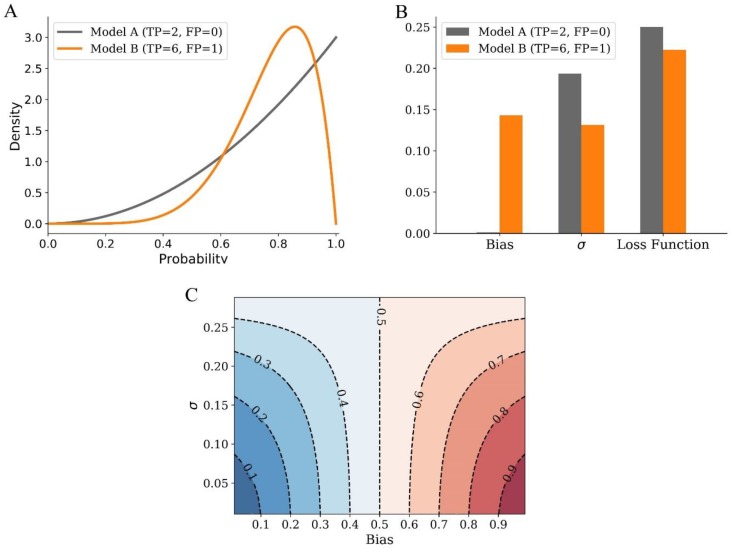
Bias and variance analysis of LBS (Local Beta Screening). (**A**) Probability density of screening model A (TP = 2 and FP = 0) and B (TP = 6 and FP = 1). (**B**) Comparison of screening model A and B by bias, variance, and loss function of LBS. Variance is shown in the form of σ (standard deviation) so that it is comparable with bias and the value of loss function, similarly hereinafter. (**C**) General relationship among bias, variance, and loss function of LBS. TP: true positive. FP: false positive.

**Figure 2 molecules-24-02414-f002:**
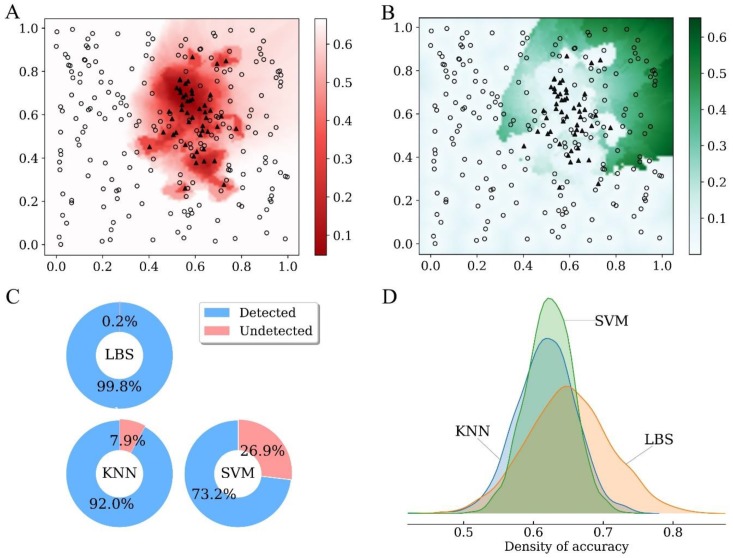
Performance of LBS in a simulation study. (**A**) Value of loss function for all points in raw space without feature noise. Circle, sample of negative class; Triangle, sample of positive class. (**B**) Radius of LBS for all points in raw space without feature noise. Circle, sample of negative class; Triangle, sample of positive class. (**C**) Percentage of noisy features detected by different screening approaches. (**D**) Comparison of screening accuracy on test sets. Density of accuracy was estimated on 2000 independent test sets, each of which corresponds with a trained model. KNN: K-nearest neighbors. SVM: support vector machine.

**Figure 3 molecules-24-02414-f003:**
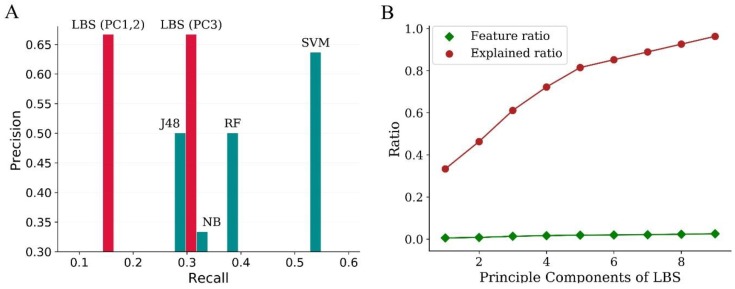
Comparison of LBS to other machine learning algorithms on dataset of inhibitors of Rho kinase 2. (**A**) Comparison of LBS to the four machine learning algorithms described by Schierz. (**B**) Relationship of feature ratio and sample ratio to principle components of LBS. NB: naive Bayes. RF: random forest. J48: J48 version of decision tree. PC: principle component.

**Figure 4 molecules-24-02414-f004:**
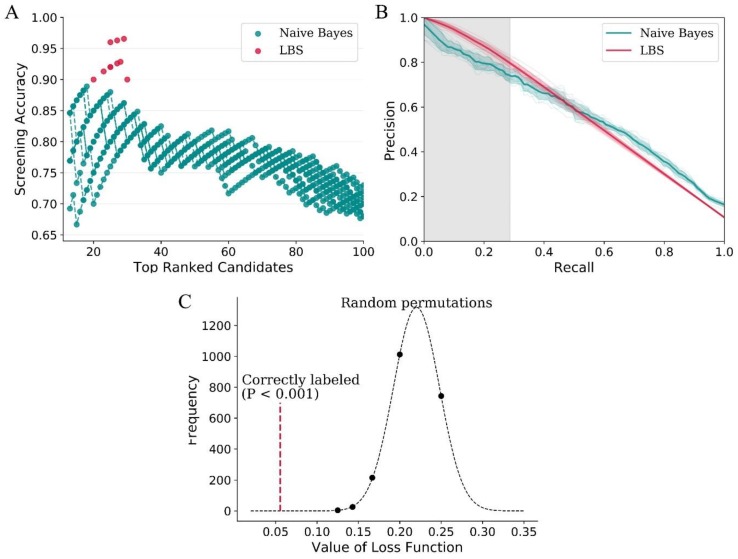
Performance of LBS on data of compound screening. (**A**) Screening accuracy of LBS and NB. (**B**) Precision–recall curve for LBS and NB. The gray-filled part was the screened zone in PC1 of LBS. The AUC (area under curve) of LBS in the screened zone was 0.267 ± 0.004, and the corresponding value of NB was 0.246 ± 0.005. (**C**) Permutation test for LBS. The values of dependent variable were randomly permutated 2000 times to calculate the loss values of LBS. The dashed red line represents the loss value of LBS obtained from the actual experiment (*p* = 1.8e^−9^).

**Figure 5 molecules-24-02414-f005:**
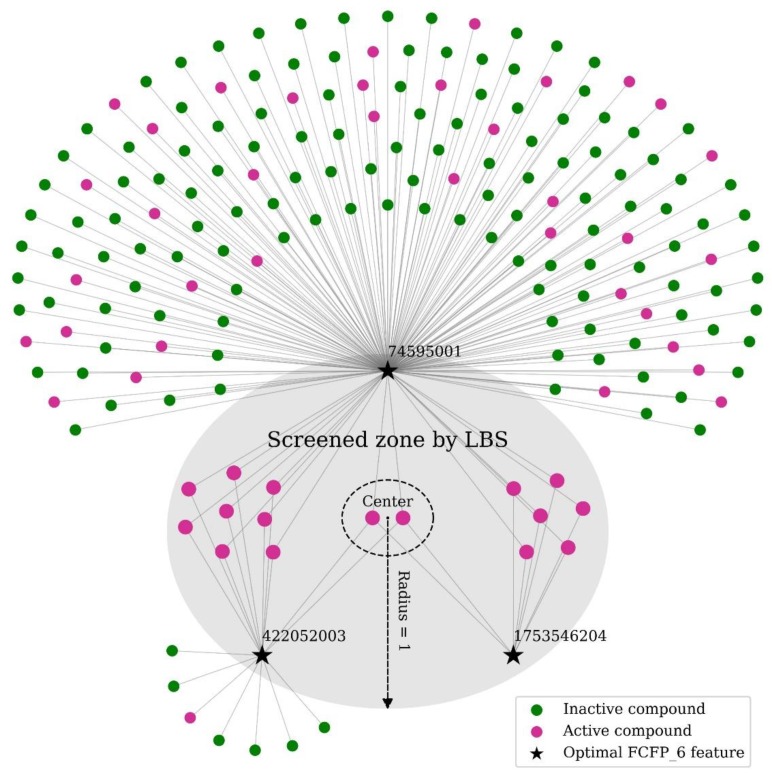
Interpretation of the model trained by LBS. A node represents a compound or a feature, and an edge means that the feature exists in the structure of the corresponding compound. In the optimal screened zone (gray-filled part) trained by LBS, the center (black dashed circle) connected to all the three optimal FCFP_6 features and the radius was one. It can be interpreted as a screening rule that a candidate is regarded as an active compound if its structure contains at least two of the three optimal features. FCFP: functional-class fingerprints.

**Figure 6 molecules-24-02414-f006:**
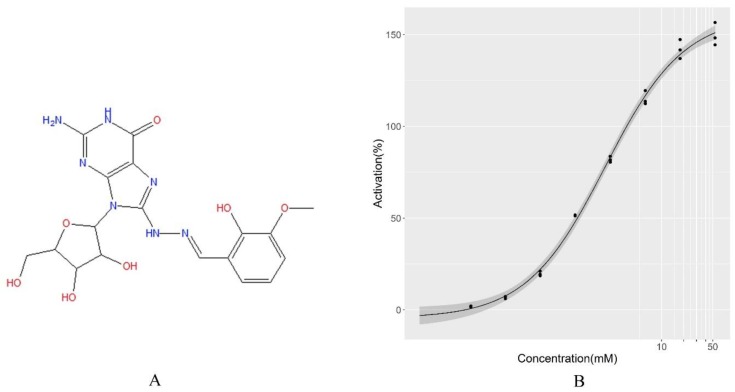
Chemical information of MCULE-2245265974. (**A**) Chemical structure of MCULE-2245265974. (**B**) Dose–response curve of MCULE-2245265974. The corresponding value of EC_50_ is 0.71 µM.

**Figure 7 molecules-24-02414-f007:**
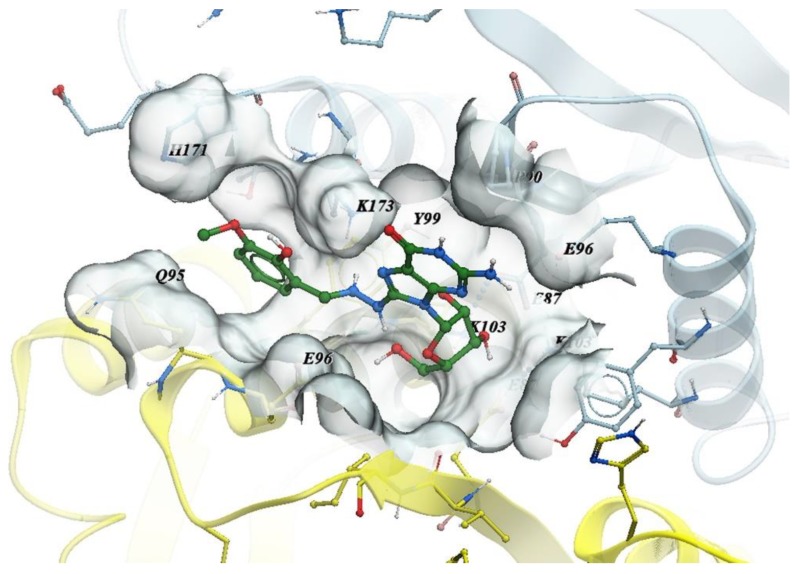
Binding mode of MCULE-2245265974 to HIV-1 integrase. The structure of two HIV-1 integrase monomers is shown in cartoon with chain A colored light cyan and chain B colored light yellow. Color coding of MCULE-2245265974: green carbon, red oxygen, blue nitrogen, gray hydrogen.

**Figure 8 molecules-24-02414-f008:**
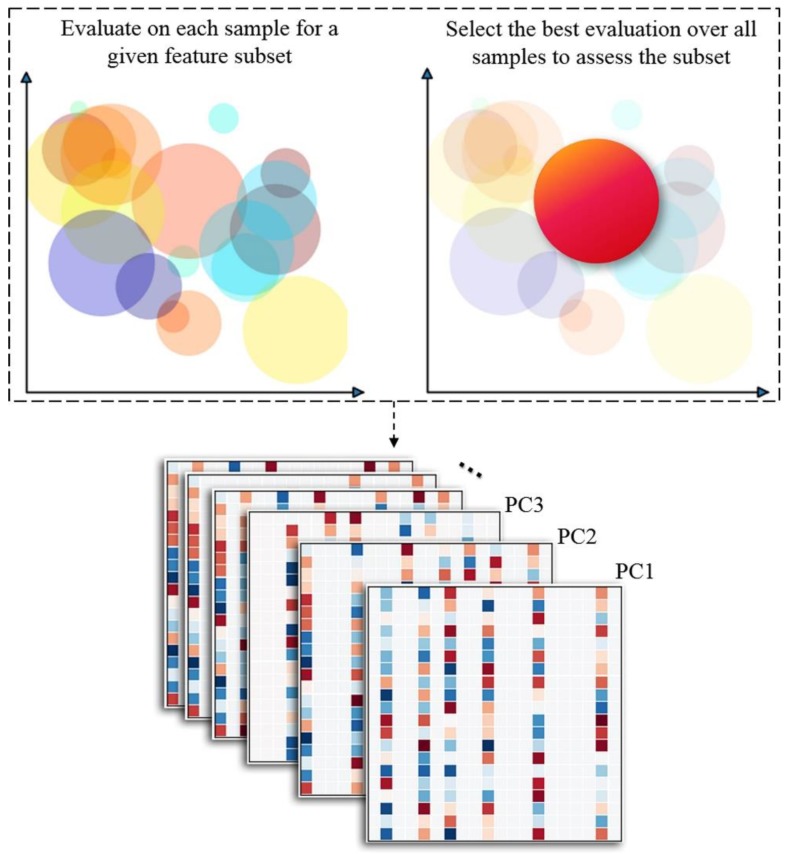
Description of LBS. For a given feature subset during training, LBS evaluates the loss function on each sample and optimizes the radius automatically. The best, rather than the summation, of evaluations is then selected as metric of assessing the feature subset. The final LBS model consists of multiple PCs, and each PC is a sparse combination of features.

**Table 1 molecules-24-02414-t001:** Compounds proved to be active in binding assay for validation of the screening results by LBS. A compound is regarded as active when the corresponding EC_50_ < 5 µM. More details can be found in Appendix A.

Pubchem CID	Molecular Formula	Molecular Weight	EC_50_
5291488	C_21_H_20_O_11_	448.38	3.62
5378597	C_21_H_20_O_12_	464.39	1.36
135427812	C_18_H_21_N_7_O_7_	447.41	0.71
3656702	C_14_H_11_FN_2_O_4_	290.25	4.88
4137880	C_13_H_18_N_2_O_7_	314.30	1.26
46397461	C_19_H_18_O_12_	438.35	1.26

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
