# Peer review of "A Ligand-Based Virtual Screening Method Using Direct Quantification of Generalization Ability"

_molecules, 2019, doi:10.3390/molecules24132414_

Round 1

Reviewer 1 Report

The manuscript entitled "A Ligand-Baed Virtual Screening Method Using Direct Quantification of Generalization Ability" reports an interesting study about a machine learning algorithm LBS (Local Beta Screening) used to perform a ligand-based virtual screening to select potential active compounds activators of HIV-1 integrase multimerization in an independent compound library. The manuscript is interesting since the authors evaluate the performance of a machine learning algorithm to select compounds using high-dimensional data and they validated the results.

Reviewer 2 Report

The authors developed new machine learning approach for the ligand-based virtual screening applications. The methodology was verified both in in silico studies, but in addition, the application in virtual screening campaign was carried out and several active compounds were identified thanks to this procedure. However, there are several issues that need to be improved before the paper is ready for publication.

The resolution of figures in too low, moreover, the font on axes and legends on the figures should be of bigger size.

Other measures (than recall and precision, such as balanced accuracy, auc, mcc) should be used in the retrospective validation of LBS. Moreover, each method with which the LBS is compared should be optimized (SVM in particular as it is very prone to settings). In addition, random cross-validation (as even mentioned in the introduction part) is very prone to bias and therefore it should be repeated at least 10 times and the results reported should be averaged over these 10 runs, together with the standard deviation in evaluating parameters values

In my opinion, considering 10 uM as threshold for compound activity is too high (especially in terms of EC50). I would put 1 uM, or at least 5 uM.

Author Response

Point 1: The resolution of figures is too low, moreover, the font on axes and legends on the figures should be of bigger size.

Response 1: Thanks for the suggestion. We have redrawn all the figures in the manuscript. Now the font and legends are of bigger size and the resolution of Figure1-8 has been adjusted to 600 dpi.

Point 2: Other measures (other than recall and precision) should be used in the retrospective validation of LBS.

Response 2: Thanks for the comments. We have added two new measures: balanced accuracy and Mathews correlation coefficient in the retrospective validation of LBS. “The balanced accuracy of LBS (56.3% ± 0.8%) was not significantly different from that of NB (56.4% ± 0.4%), and the results of Mathews correlation coefficient were similar (0.149 ± 0.010 and 0.147 ± 0.007 for LBS and MCC respectively).” (lines 153-155).

Point 3: Each method with which the LBS is compared should be optimized.

Response 3: Thanks for the comments. Actually, optimization for the methods with which LBS was compared had been done but was not described in the manuscript. Now we have added a description on the corresponding optimization as follows: “The parameters of models were optimized during training. For KNN, the number of nearest neighbors was optimized in the range of all odd numbers from 1 to 201. The polynomial kernel function was used in SVM. Degree of polynomial and the regularization parameter were optimized simultaneously by grid search, where the range were 1 to 5 and 500 to 1000 for degree of polynomial and the regularization parameter respectively.” (lines 93-97).

Point 4: Cross validation should be repeated at least 10 times and the results reported should be averaged over these 10 runs, together with the standard deviation in evaluating parameters values.

Response 4: Thanks for the suggestion. We have repeated the cross validation 10 times and the corresponding results have been updated in this revised version. In Figure 4A, all the results of repeated 10 runs of screening accuracy were shown individually. In Figure 4B, the standard deviations of PR curve were shown as red-filled and cyan-filled areas for LBS and NB respectively. The value was updated on line 145, which is “88.9%”. The standard deviations were added on line 146, 151, 152 and 153 respectively.

Point 5: Considering 10 μM as threshold for compound activity is too high.

Response 5: Thanks for the comments. We now considered 5 μM as the threshold in the revised manuscript. Table 1 was correspondingly updated. Line 22 “6 were prove to be active” and line 193 “6 were prove to be active” were updated.

Round 2

Reviewer 2 Report

The paper can be published in the current form